

# Acoustic mapping of mixed layer depth

Christian Stranne[1,2], Larry Mayer[3], Martin Jakobsson[1,2], Elizabeth Weidner[3], Kevin Jerram[3], Thomas C. Weber[3], Leif G. Anderson[4], Johan Nilsson[2,5], Göran Björk[4], Katarina Gårdfeldt[6]

[1]Department of Geological Sciences, Stockholm University, Stockholm, Sweden
[2]Bolin Center for Climate Research, Stockholm University, Stockholm, Sweden
[3]Center for Coastal and Ocean Mapping, University of New Hampshire, Durham, New Hampshire, USA
[4]Department of Marine Sciences, University of Gothenburg, Gothenburg, 40530, Sweden
[5]Department of Meteorology, Stockholm University, Stockholm, Sweden
[6]Department of Chemistry and Chemical Engineering, Chalmers University of Technology, Göteborg, Sweden

**Abstract.** The ocean surface mixed layer is a nearly universal feature of the world oceans. The depth of the mixed layer (MLD) influences the exchange of heat and gases between the atmosphere and the ocean and constitutes one of the major factors controlling ocean primary production as it affects the vertical distribution of biological and chemical components in near-surface waters. Direct observations of the MLD are traditionally made by means of conductivity, temperature and depth (CTD) casts. However, CTD instrument deployment limits the observation of temporal and spatial variability of the MLD. Here, we present an alternative method where acoustic mapping of the MLD is done remotely by means of commercially available ship-mounted echosounders. The method is shown to be highly accurate when the MLD is well defined and biological scattering does not dominate the acoustic returns. These prerequisites are often met in the open ocean and it is shown that the method is successful in 95% of data collected in the central Arctic Ocean. The primary advantages of acoustically mapping the MLD over CTD measurements are: (1) considerably higher temporal and horizontal resolutions and (2) potentially larger spatial coverage.

## 1 Introduction

The surface mixed layer is an important and nearly universal feature of the world oceans. It is defined as a quasi-homogeneous layer that extends from the surface down to the penetration depth of turbulent mixing, generated by wind stress and buoyancy fluxes at the air-sea interface (Kraus & Turner, 1967; Price et al., 1986). The MLD is an important parameter within several atmospheric and oceanographic research disciplines as the transfer of mass, momentum, and heat across the mixed layer provides the source of almost all oceanic motions (de Boyer Montegut et al., 2004). Variations in MLD influence air-sea interactions through the storage of various gases, such as carbon dioxide and methane (Kraus & Businger, 1994). The MLD also affects the vertical distributions of dissolved and particulate biological and chemical components in surface waters (Gardner et al., 1995), and is thus one of the main factors controlling the primary production (Behrenfeld & Falkowski, 1997; Sverdrup, 1953). The MLD is also of importance since it represents a reservoir for pollutants that are deposited from the atmosphere and cycled between the atmosphere and the surface waters (Nerentorp Mastromonaco et al., 2017). Furthermore, temporal and spatial variability in the MLD is essential for validating and improving mixed layer parameterizations (Ling et al., 2015; Martin, 1985; Noh et al., 2002), and as diagnostics in mixed layer budgets (Hasson et al., 2013; Montégut et al., 2007). Its depth, properties and behavior also play an important role in understanding acoustic propagation in the ocean.

The MLD is controlled primarily by surface stress (exerted by wind or sea-ice), buoyancy fluxes (heating/cooling, ice melt/formation, or precipitation/evaporation), and dissipation (Large et al., 1994). Thus, any variation in the MLD can be linked to these processes. It is well established that the MLD varies on diurnal to inter-decadal timescales (Bissett et al., 1994; Kara et al., 2003; Li et al., 2005; Polovina et al., 1995), but higher frequency variability is poorly understood due to observational limitations. For direct measurements of the MLD, various forms of conductivity, temperature, depth (CTD) sensor data are collected from ships, moorings, or gliders. These collect discrete profiles through the water column with a temporal sampling frequency of typically less than one profile per 10 minutes. Broad global coverage of the distribution of the MLD is becoming increasingly available through salinity and temperature stratification data from the ARGO float program (Freeland et al., 2010), but the high spatial frequency of ocean thermohaline variability is still strongly undersampled (Guinehut et al., 2012). Satellite-derived products provide global synoptic coverage of, for example, sea level (MacIntosh et al., 2016), sea



surface temperature (Donlon et al., 2009) and sea surface salinity (Font et al., 2013; Lagerloef et al., 2012), but are restricted to sea surface properties.

Since the early 20[th] century, active acoustic sensors have been used to track military targets in the water column (MacLennan & Simmonds, 2013). Not long after the first military applications, acoustic water column mapping with echosounders was applied to fisheries science, where detection and quantification of fish distributions were the primary focus (Kimura, 1929; MacLennan, 1990). The applications of acoustic water column mapping have broadened in recent years to include marine ecosystem acoustics (Benoit-Bird & Lawson, 2016; Godø et al., 2014), observations of gas bubbles and oil droplets associated with natural seeps (Jerram et al., 2015; Merewether et al., 1985), and fossil fuel production (Hickman et al., 2012; Weber et al., 2012). Acoustic imaging of the water column has also been used within the field of physical oceanography; single beam echosounders can capture fine-scale oceanographic structures, typically attributed to biological scatters or turbulent microstructures (Klymak & Moum, 2003; Pingree & Mardell, 1985; Trevorrow, 1998). Larger scale thermohaline structures have been observed with lower frequency seismic systems (e.g., Holbrook et al., 2003). Custom-built echosounders utilizing wideband frequency-modulated pulses have been deployed since the 1970s (e.g. Holliday, 1972), but have received renewed attention as they have become commercially available (Duda et al., 2016; Lavery et al., 2010; Stranne et al., 2017). Advantages of wideband echosounders, compared to conventional narrow-band systems, include increased signal-to-noise ratio (SNR) and increased range resolution through pulse-compression processing (Stanton & Chu, 2008; Turin, 1960), and the ability to study the frequency response of individual targets (Lavery et al., 2010; Stanton et al., 2010).

The increased SNR of wideband echosounders have made it possible to map density stratification in the ocean. Stranne et al. (2017) were able to acoustically image individual thermohaline steps resulting from the intrusion of warm and salty Atlantic waters into the colder and less saline Arctic waters. The range resolution provided by the wideband sonar enabled the detection of individual density layers separated by less than 0.5 m to depths of about 300 m. These thermohaline layers represent change in temperature of typically 0.05 °C and change in salinity of 0.015, with corresponding acoustic reflection coefficients at the layer interface as low as $2 \cdot 10^{-5}$. Although the ensonified area is smaller at shallower depths (leading to a weaker scatter strength) this is compensated by generally higher reflection coefficients at the base of the mixed layer, meaning that the MLD can be detectable with wideband echosounders. Here we show that underway profiling using wideband echo sounding systems at up to several pings per second can map the behavior of the MLD at very high spatial resolution.





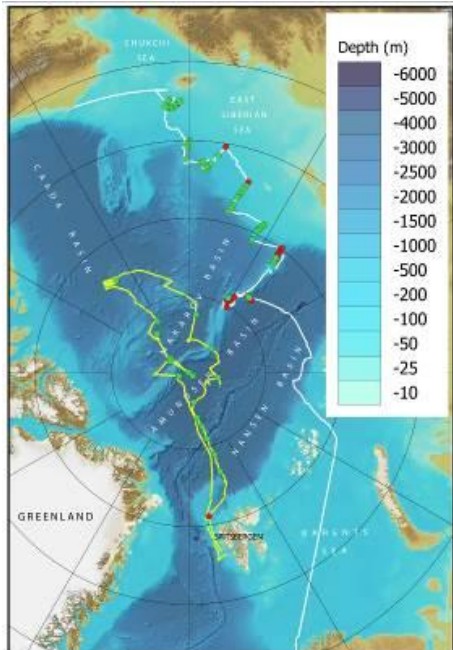

**Figure 1. Map showing cruise tracks for the SWERUS-C3 cruise (white) and the Arctic Ocean 2016 cruise (yellow). CTD stations are shown as dots where green indicates the MLD was successfully observed acoustically, red indicates the MLD was not successfully observed acoustically, and blue indicates no mixed layer was present.**

## 2 Methods

### 2.1 Data and the regional setting

10 Acoustic water column data were collected throughout the Arctic Ocean during two expeditions with Swedish icebreaker (IB) *Oden*; Leg 2 of the Swedish-Russian-US Arctic Ocean Investigation of Climate-Cryosphere-Carbon Interactions 2014 Expedition (SWERUS-C3) and the Arctic Ocean 2016 Expedition (AO2016).

Leg 2 of SWERUS-C3 departed 20 August 2014 from Barrow, Alaska, and ended 4 October in Tromsø, Norway. The expedition covered mainly the shallow areas of the East Siberian Sea continental shelf and shelf slope (Fig. 1). The median water depth of the 78 CTD stations investigated from SWERUS-C3 is 340 m. The hydrography of 15 this area can be characterized as dynamic and seasonally variable as it is influenced by large river runoff, coastally trapped waves, ice formation and melting, and brine rejection in coastal polynyas.

The AO2016 expedition took place between 8 August and 19 September, 2016, departing from and returning to Svalbard (Fig. 1). One specific research goal during AO2016 was dedicated to investigating the possibility to detect and map thermohaline stratification using a midwater wideband echosounder. The cruise track covered 20 mainly the central Arctic Ocean and the median water depth of 24 CTD stations investigated is 4000 m (Fig. 1). Together, the SWERUS-C3 and AO2016 expeditions spanned the breadth and depth of the Arctic Ocean and provided wideband acoustic data in a variety of oceanographic settings.

### 2.2 Wideband water column acoustic data collection

25 The wideband water column backscatter data presented here were collected with a Simrad EK80 split-beam scientific echosounder (SBES) installed in IB *Oden*. The system was operated continuously during both the SWERUS-C3 and AO2016 expeditions.



The SBES consisted of a Simrad EK80 wideband transceiver transmitting through a standard Simrad ES18-11 transducer installed in the 'ice knife' near the bow of the vessel and protected by an ice window. This transducer model is widely installed in fishery research vessels, typically operating at 18 kHz with a -3 dB beamwidth of 11°. In 2014, the transducer model was tested with a Simrad EK80 wideband transceiver and determined to have a
useable two-way frequency response over 15-25 kHz. Thus, a frequency range of 15-25 kHz was used throughout the EK80 data collection period on IB *Oden*.

Transmit power was maintained at the maximum setting of 2000 W to compensate for losses through the ice protection window and improve signal-to-noise (SNR) characteristics, especially during noisy hull-ice interactions. Transmission pulse lengths were adjusted over a range of 1-8 ms, in an effort to minimize the extent
of autocorrelation sidelobes (sidelobes are typically minimized with shorter pulses) while maximizing the SNR (better with longer pulses). All EK80 operation was controlled and monitored around-the-clock using the Simrad user interface to adjust pulse length and range recording duration. Data were logged in the Simrad .raw format.

Position and attitude information were provided to the echosounder as an integrated solution by a Seapath Seatex 330 GPS/GLONASS navigation and motion reference system. Vessel motion was minimal (typically less than 1°
pitch and roll, in the data presented here) and thus does not appreciably affect the observations of horizontally-oriented backscattering layers occupying broad portions of the beam.

During the AO2016 expedition, a small delay was applied to the EK80 transmit-receive cycle trigger in order to avoid transmission interference from the two other echo sounding systems (Kongsberg EM122 12 kHz multibeam and SBP120 2-7 kHz sub-bottom profiler) in the earliest portion of the EK80 receive cycle, corresponding to the
upper water column region of interest.

### 2.3 EK80 post processing methodology

The dataset collected with the EK80 was match-filtered with an ideal replica signal using a MATLAB software package provided by the system manufacturer, Kongsberg Maritime (Lars Anderson, personal communication).
After match-filtering, ship-related noise was found within the signal band. A bandpass filter with 16 and 22 kHz cut-off frequencies was applied to the data to exclude the noise. Sound speed profiles were calculated from CTD-derived temperature, salinity and pressure data using the International Thermodynamic Equation of Seawater (Commission et al., 2010). Ranges from the transducer were then calculated using the cumulative travel times through sound speed profile layers based on the nearest (in time) CTD profile. These ranges were then converted
to depths by compensating for the transducer location relative to the static waterline on IB *Oden* and the heave of the vessel.

### 2.4 EK80 extended target calibration procedure

The EK80 was calibrated onboard the *Oden* on 1 September 2015, following a standard methodology described
by (Demer et al., 2015). A 64 mm copper sphere of known acoustic properties was suspended on a monofilament line and moved through the SBES field of view. The calibration data were collected in relatively calm seas and atmospheric conditions while the *Oden* drifted. All propulsion systems were secured during the calibration procedure in order to reduce noise in the water column. A CTD was collected immediately before calibration operations.

Utilizing a calibration sphere target strength model based on the work by Faran (1951) and MacLennan (1981) (MATLAB software package available at www.ices.dk), a calibration offset ($C = 8.5$ dB, averaged over the transducer beam width) was calculated using a temperature of 0 °C and a salinity of 34.5 at the sphere depth of approximately 80 m. This calibration offset represents the difference between the nominal target strength (*TS*) observed by the EK80, as predicted after match filtering, and the modeled *TS* of the calibration sphere. The offset
is then applied to subsequent measurements of *TS*, yielding calibrated *TS* results for the EK80 datasets.

### 2.5 Estimates of the reflection coefficient from EK80 observations

The *TS* of an ideally smooth layer is a function of both the reflection coefficient (*R*), and the ensonified area (*A*). Here, we assume that *A* is limited by the width of the EK80 beam (rather than the length of the pulse), such that *A* can be estimated as



$$A(z) = \pi (\tan(\varphi)\, z)^2 \ ,$$

where φ is half the beam width and $z$ is the depth in the sonar reference frame. Following (Lurton & Leviandier, 2010) the *TS* for a layer at depth $z$, with reflection coefficient $R$, can then be estimated as

$$TS(z) = 20 log_{10} R + 10 log_{10}(A(z)) \ .$$

For our estimates of observed *R*, we simply invert the above equation to solve for *R*:

$$R = 10^{(TS - 10 log_{10} A)/20} \ ,$$

where *TS* is the calibrated acoustic backscatter observation from the EK80.

**2.6 CTD**

CTD data were collected with a SeaBird 911 equipped with dual SeaBird temperature (SBE 3) and conductivity (SBE 04C) sensors. The CTD data files were post processed with SBE Data Processing software, version 7.26.6 (available at www.seabird.com). The alignment parameter was tuned following the suggested method described in the SBE Data Processing manual (available at www.seabird.com). All CTD data presented are averaged in 10 cm vertical bins.

The reflection coefficient from CTD data ($R_{CTD}$) was calculated through

$$R_{CTD}(i) = \frac{\eta(i) - \eta(i-1)}{\eta(i) + \eta(i-1)} \ ,$$

where each element *i* has a corresponding depth $z(i)$, the depth of $R_{CTD}(i)$ is the average of $z(i-1)$ and $z(i)$, and η is the acoustic impedance given by

$$\eta(z) = V(z)\rho(z) \ ,$$

where *V* is the sound speed and *ρ* the seawater density. The accuracies of the pressure, conductivity and temperature sensors are 0.0015%, 0.0003 S/m and 0.001 °C, respectively (www.seabird.com). All conversions (salinity, density and sound speed) were made according to the International Thermodynamic Equation of Seawater (Commission et al., 2010).

**2.7 MLD derived from CTD**

To determine the MLD, we apply the method presented in de Boyer Montégut et al. (2004), where successively deeper data points in each of the CTD potential temperature profiles are examined until one is found with a potential temperature value differing from the value at the 10 m reference depth by more than the threshold value (ΔT) of ±0.2 °C. Using this approach, the MLD is then assumed to be at least 10 m deep, and any shallower well-
mixed sections in the water column are not taken into consideration (de Boyer Montegut et al., 2004).

**3 Results**

We investigated the shallow (<50 m depth) EK80 water column data from approximately one hour before to one hour after the time of each CTD cast, for a total of 102 CTD stations throughout both expeditions (Fig. 1). An
example of acoustic mapping of the MLD over a 117 km long cruise track (about 12 hours) in the central Arctic Ocean is shown in Figure 2.




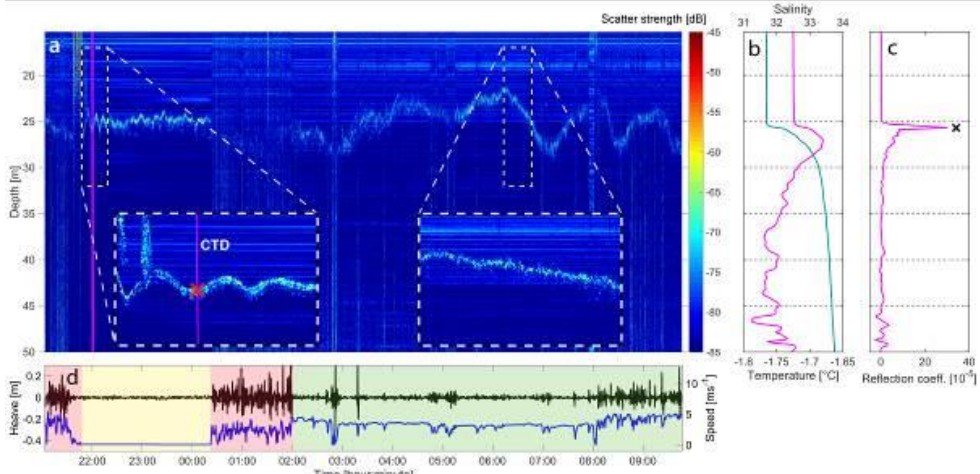

**Figure 2. Continuous tracking of MLD in central Arctic Ocean over a 117 km cruise track. (a) EK80 echogram (2 ms pulse length) with magnified insets (dashed boxes) showing the MLD while drifting (left) and while steaming (right). (b) CTD profiles showing temperature (magenta) and salinity (cyan). (c) reflection coefficients derived from CTD data (magenta) and from scatter strength (black cross represents the observed scatter strength of -65 dB at this depth extracted from the left inset in a). (d) heave (black), speed over ground (blue), and time periods corresponding to ice breaking (red), steaming (green), and drifting (yellow). Vertical magenta lines in a show the position of the CTD. The red cross in a (left inset) marks the depth of the reflection coefficient spike in c. Note that the ability to detect MLD acoustically is severely reduced while breaking ice.**

We categorize CTD stations where EK80 data is available into three classes (Fig. 1): green indicates a mixed layer is present in the CTD data and the MLD is visible in the EK80 data (success); red indicates a mixed layer is present in the CTD data but the MLD is not visible in the EK80 data (failure); and blue indicates a mixed layer is not present in the CTD data. The classification is done subjectively by visual scrutiny of each echogram and subsequent comparison with CTD profiles; this process is meant to provide a general idea of how often a mixed layer is present in the *in situ* CTD data and the success rate of the remote EK80 MLD detection. In order to automate the EK80 MLD detection process, a stratification tracking tool needs to be produced. No such tool is available but methods used within the seismic processing or seismic oceanography fields can likely be applied also to sonar data.

Of the 102 CTD stations investigated, a mixed layer is present in 91 CTD profiles (90 %); of these 91 confirmed MLD profiles, the MLD is simultaneously visible in the EK80 data in 69 instances (76 %) (Table 1). The $\Delta T$ threshold estimate method yielded similar results to that of using acoustic data, with a root-mean-square deviation (rmsd) of about 3 m (Table 2). The original $\Delta T$ threshold (0.2 °C) as presented in de Boyer Montégut et al. (2004) worked well for the SWERUS-C3 CTD stations but generally failed in the central Arctic Ocean (as shown in Fig. S3). Therefore, we used a modified $\Delta T$ threshold of 0.05 °C on CTD data from AO2016. Note that, even though instances where the $\Delta T$ threshold method clearly fails are excluded in these statistics, there are still instances where it provides less than ideal MLD estimates. The deviation therefore reflects inaccuracies in both methods.

**Table 1. Success and failure rates of acoustic detection of MLD when present in CTD data.**

| Category of detection | SWERUS-C3 | AO2016 | Total[*] |
|---|---|---|---|
| MLD present in CTD profile | 69 | 22 | 91 |
| MLD in CTD and in EK80 (success) | 48 (70 %) | 21 (95 %) | 69 (76 %) |
| MLD in CTD but not in EK80 (failure) | 21[**] (30 %) | 1 (5 %) | 22 (24 %) |





*Of the total 102 CTD stations investigated, 11 stations (9 in SWERUS-C3 and 2 in AO2016) did not have a well-defined MLD (blue category in Fig. 1) and are not included in these statistics. An example of this category is shown in Fig. S1.

** More than half of the 21 acoustic detection failures in the SWERUS-C3 data are related to the relatively large
5     ship draft of IB *Oden*, and four of the failures are related to noise of unknown source that appeared in the EK80 data towards the end of the cruise. When not counting these stations, the MLD acoustic detection success rate is close to 90% in the SWERUS-C3 data.

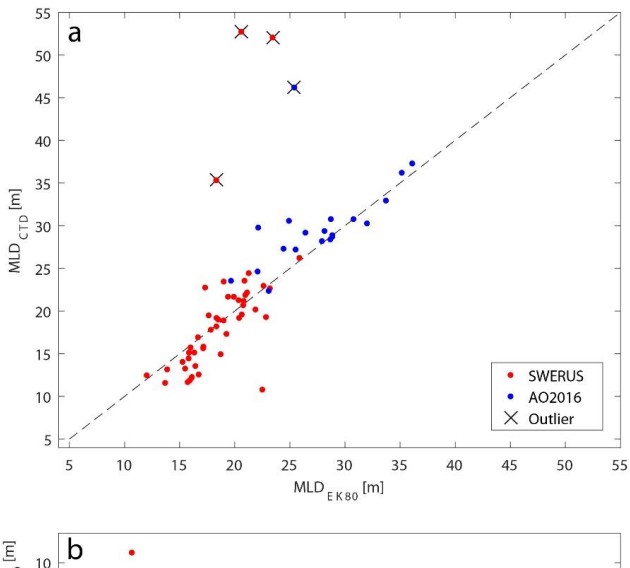

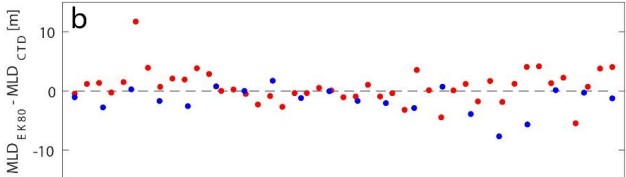

**Figure 3. (a) MLD for the individual stations derived from CTD (MLD$_{CTD}$) versus MLD derived from EK80 data**
10     **(MLD$_{EK80}$). (b) Difference between MLD$_{EK80}$ and MLD$_{CTD}$. In total, four outliers (black crosses in (a)) where the ΔT threshold method fails (as exemplified in Figure S2) are excluded from the statistics. Note that the original ΔT threshold (0.2 °C) as presented in de Boyer Montégut et al. (2004) generally failed in the central Arctic Ocean (Figure S3) and that we instead used a modified ΔT threshold of 0.05 °C on CTD data from AO2016.**

15     **Table 2. Statistics for MLD$_{EK80}$ and MLD$_{CTD}$ with the four outliers (Figure 3) excluded; all units are meters.**

| MLD (m) | mean MLD$_{CTD}$ | mean MLD$_{EK80}$ | std MLD$_{CTD}$ | std MLD$_{EK80}$ | rmsd |
|---|---|---|---|---|---|
| SWERUS-C3 | 17.7 | 18.6 | 4.2 | 2.9 | 2.8 |
| AO2016 | 29.2 | 27.5 | 3.6 | 4.3 | 2.7 |
| ALL MLD DETECTIONS | 21.3 | 21.3 | 6.7 | 5.3 | 2.8 |



## 4 Discussion

### 4.1 MLD observations

The typical summer MLD of the Arctic Ocean is ∼20 m (Steele et al., 2008). Toole et al. (2010) reported, for the central Canada Basin, an average summer MLD of 16 m and an average winter MLD of 24 m. The shallower mean
MLD in the SWERUS-C3 data is consistent with the large river runoff into the Siberian shelf seas, which should lead to a generally shallower mixed layer compared to the AO2016 data from the central parts of the Arctic Ocean (Large et al., 1994). Given the dynamic nature of the more coastal Leg 2 SWERUS-C3 cruise track compared to the open ocean-dominated AO2016 cruise track, we were expecting larger MLD variability in the SWERUS-C3 data. We cannot see such tendency in our data (Table 2), but again the basis of the statistics is rather poor.

In general, MLD variations between the different regions of the Arctic Ocean covered in this study match well with mean Arctic Ocean MLD based on other field observations (Ilıcak et al., 2016; Peralta-Ferriz & Woodgate, 2015), with shallow MLDs along the East Siberian Sea, slightly deeper MLDs in the Canada Basin and deepest MLDs in the central Arctic Ocean. As the emphasis of this paper is mainly on the acoustic method rather than the actual MLD observations, we are hesitant to draw any conclusions based on the MLD statistics presented in Table
2, especially when considering the small number of observations on which the statistics are based on.

### 4.2 Sampling frequency

With the acoustic method we can observe the MLD at a horizontal resolution far exceeding alternative *in situ* methods, such as CTD profiles. The acoustic method enables the study of internal waves propagating on the layer
interface at the base of the mixed layer (left inset, Fig. 2a). The recording duration of the EK80 was set to observe the full water column, resulting in a ping frequency of around 0.1 ping s⁻¹ in deep water, though the ping rate can be set much higher (up to several pings per second) in shallow water or if only data from the shallow part of the water column are to be collected. In our data the MLD is clearly visible while drifting and steaming, but the quality of the data underway would benefit from a higher ping rate; specifically, the highest-frequency temporal and/or
spatial variations in MLD are likely undersampled at this lower ping rate while the vessel is moving (right inset Fig. 2a).

### 4.3 Vertical detection limits

The cruise track of the SWERUS-C3 expedition during Leg 2 covers mainly the shallow areas of the East Siberian
Sea shelf and shelf slope, an area that is heavily influenced by river runoff (e.g., from the Lena River). The freshwater input (or negative buoyancy flux) to the coastal waters leads to generally shallower MLD (Large et al., 1994). This is clearly manifested in our data where the average MLD from the shelf-dominated SWERUS-C3 cruise is more than 60 % shallower than that of the sea ice-covered, deep basin-dominated AO2016 cruise (Table 2).

The deep depth limit for detecting ocean stratification with this particular EK80 setup appears to be around 300 m (Stranne et al., 2017) while the shallow depth limit depends on the draft of the hull-mounted transducer and the pulse length. On the *Oden,* the EK80 transducer is mounted at a draft of 7 m and, depending on pulse length, we generally observe useful data starting at 7.5-12 m depth from the surface (0.5-5 m from the transducer, Fig. 4d). The amount of data lost at the upper boundary is reduced with shorter pulse length (Fig. 4d); these data also show
the better range resolution obtained with shorter pulse length (Fig. 5), but there is a serious tradeoff in terms of reduced SNR (Fig. 4a). More data are needed in order to determine the optimal pulse length for EK80 MLD detection as it also depends on region and platform.

Due to ship draft and the data loss at very close range from the transducer, the shallow MLDs seen in some of the SWERUS-C3 CTD profiles are sometimes difficult to detect acoustically with the EK80 (Figure S4). This is the
most common factor explaining more than 50% of the failures to acoustically detect the MLD during SWERUS-C3.



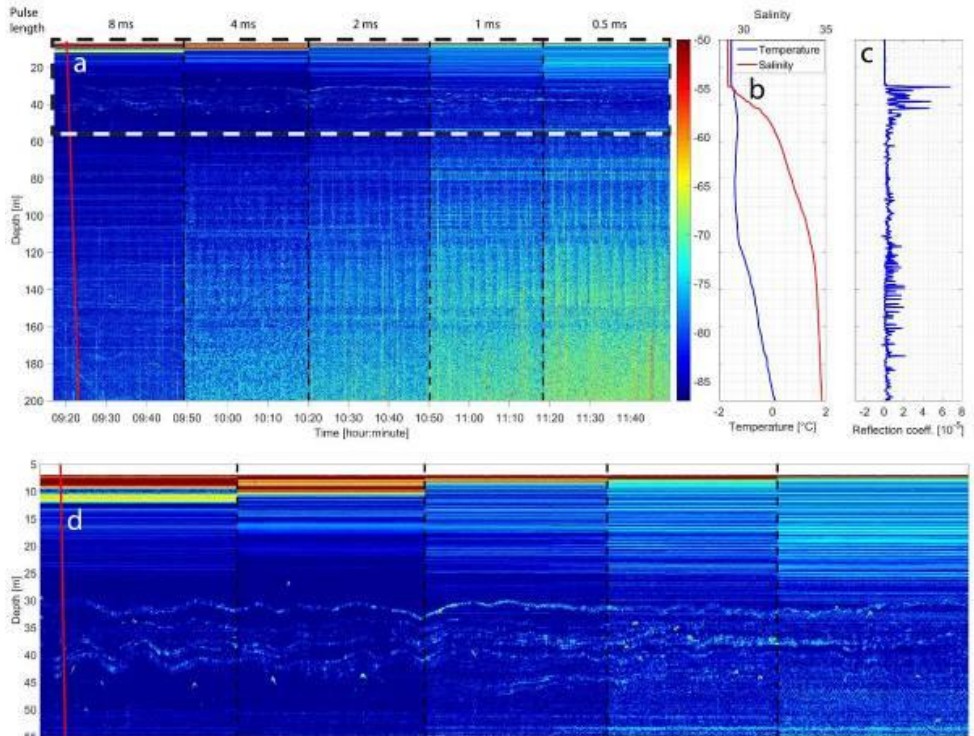

**Figure 4. Comparison of EK80 data with different pulse lengths. (a) EK80 echogram. (b) CTD profiles showing temperature (blue) and salinity (red). (c) Reflection coefficients derived from CTD data. (d) Enlargement of dashed box in a. In a and d, the vertical red line is the CTD position and the vertical dashed black lines indicate changes in pulse length (decreasing from 8 ms to 0.5 ms).**

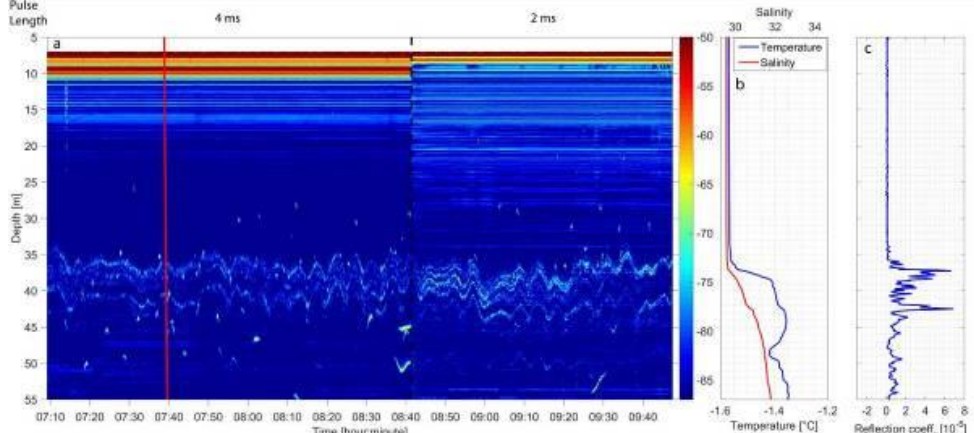

**Figure 5. Tendency of increased range resolution in EK80 data with smaller pulse length. (a) EK80 echogram with backscatter strength in dB on the color bar. (b) CTD profiles showing temperature (blue) and salinity (red). (c) Reflection coefficients derived from CTD data. Note that, as there is no ground truth CTD cast within the later section of the echogram, there might be splitting/merging of layers (as shown in Stranne et al., 2017) and other changes in the stratification behavior occurring near the change in pulse length.**



### 4.4 Biological scatter

In the example shown in Figure 2, the reflections are likely stemming from the impedance contrast from the ocean stratification alone; this is supported by the close match between the theoretical reflection coefficient calculated from the CTD data and the reflection coefficient derived from the calibrated acoustic backscatter data. This
agreement among reflection coefficients is consistent with observations of deeper thermohaline staircase stratification from the central Arctic Ocean presented in Stranne et al. (2017). In the SWERUS-C3 data, biological scatterers are generally identified at CTD stations closer to the coast. Biological scattering can potentially obscure the reflections from MLD boundary (Fig. 6); at other times, the distribution of biological scatterers may coincide with the ocean stratification and enhance the layer reflections.

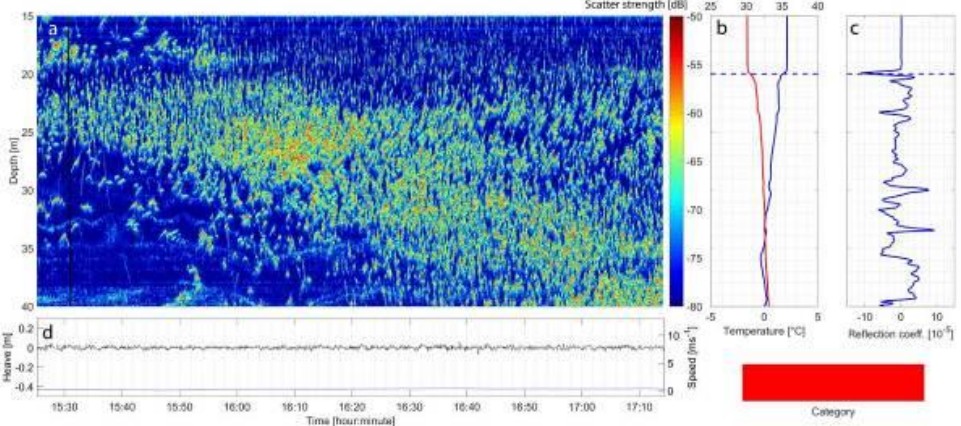

**Figure 6. MLD obscured by biological scatter. (a) EK80 echogram with black vertical line indicating the position of the CTD rosette. (b) CTD profiles showing temperature (blue) and salinity (red). (c) Reflection coefficients derived from CTD data. The horizontal dashed line in b and c show the MLD as defined by the ΔT threshold method. Also shown at the lower right is the category (red) of this particular CTD station, indicating failure of the acoustic method to detect**
**the MLD amidst strong biological scattering that spans across the MLD.**

### 4.5 Further aspects

At the time of the SWERUS-C3 expedition, we did not yet realize that the EK80 was capable of MLD detection and, accordingly, nothing was done to optimize the performance of the EK80 to detect ocean stratification in 2014.
At four of the SWERUS-C3 CTD stations, the MLD is obscured by noise from an unknown source (Fig. S5) but the source was not identified and no actions were taken to reduce it. This type of noise did not occur in the acoustic data from the later AO2016 cruise.

### 5 Conclusion

In this study we show that the MLD can be tracked acoustically, with high horizontal and vertical resolutions, over large distances (Fig. 2). The method is better suited for MLD tracking in the open ocean where it was successfully detected at 95% of the ground truth CTD stations, compared to coastal areas where the success rate was 70%. The lower success rate in coastal areas is related to the generally shallower MLDs which were sometimes impossible to detect acoustically due to IB *Oden's* vessel draft of 7 m and data loss close to the transducer. Smaller coastal
vessels with shallower draft may be better suited to acoustically track the MLD in these regions.

The acoustic method of determining MLD yields results similar to the established ΔT threshold method with a root-mean-square deviation of about 3 m. There are large uncertainties associated with the ΔT threshold method and the $MLD_{EK80}$ estimates should likely provide better precision, at least under some circumstances, as exemplified in Figure S2.





While the MLD is a crucial component within the Arctic Ocean in terms of physical, chemical and biological processes (Peralta-Ferriz & Woodgate, 2015), the discrepancy between observed and modeled MLDs in the Arctic can be quite significant (Ilıcak et al., 2016). The method of observing the MLD remotely, by means of ship-mounted echosounders, allows for larger and more efficient observational coverage. It should be noted, however, that the acoustic method cannot completely replace *in-situ* measurements (partly because of the need for ground-truthing the acoustic data), but rather it presents a very powerful complementary method to 'connect the dots' at high resolution between CTD stations.

Methods of utilizing ocean reflectivity from multi-channel seismic systems to reconstruct temperature and salinity stratification in between CTD casts have been investigated (Biescas et al., 2014; Papenberg et al., 2010; Wood et al., 2008). The increased vertical resolution (from ~10 m with multi-channel seismic data to <0.5 m with wideband acoustics (Stranne et al., 2017)) facilitates the detection of much finer thermohaline structures in the water column, including the MLD, and can potentially vastly improve these methods.

Many vessels are equipped with underway sonar systems and, thus, the method presented here is a step toward collecting large amounts of ocean stratification data globally. Such large scale acoustically obtained stratification data can become a fundamental link tying discrete ARGO float profiles (Freeland et al., 2010) with large scale synoptic coverage of sea surface temperature and salinity data derived from satellites (Font et al., 2013; Lagerloef et al., 2012). Furthermore, modeling approaches for estimating MLD are often based on remote sensing data, including LIDAR data for scattering layers and satellite data for sea surface salinity, sea surface temperature, surface wind speed and sea level (Ali & Sharma, 1994; Durand et al., 2003; Hoge et al., 1988; Yan et al., 1990). High-resolution acoustic mapping of the MLD will add important inputs to these models.

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
