# Peer review of "Acoustic mapping of mixed layer depth"

_Ocean Science, 2017_

## Referee Comment (RC1) · Anonymous Referee #1 · 5 Feb 2018

As a companion to the lead author's paper published (with some different co-authors) in Nature late last year (DOI:10.1038/s41598-017-15486-3) , Stranne and colleagues present what I would term a technical-demonstration paper showing how the depth of the ocean surface mixed layer may be sensed acoustically. The demonstration is set in the Arctic using data from two icebreaker cruises that sampled in both ice-covered and open-ocean regions at various vessel speeds. While the spin is largely positive, several limitations to the technique are discussed that will constrain where and when the approach will yield scientifically useful information.

Apologies for the cumbersome terminology, but I believe it is important to distinguish between the ocean surface "mixed layer" and the "mixing layer." I consider the latter to be the span that is actively being stirred vertically at the time of observation; it can be (and is frequently) thinner than the mixed layer whose base might mark the maximum depth of turbulent stirring in the past. My sense is that the acoustic technique

presented by Stranne and colleagues preferentially identifies the base of this deeper, possibly remnant surface mixing layer (in part due to the typically larger vertical gradient at the mixed layer base and its greater depth - at least in the data sets presented that were acquired from a large-draft vessel). Either way, I believe it is important to recognize this distinction and discuss if/how each class of "mixed" layer might be observed acoustically. One reason for worrying about (weak) stratification within the surface mixed layer is its possible manifestation of restratification processes including submesoscale instabilities (see Timmermans et al., 2011, doi: 10.1175/JPO-D-11-0125.1). Indeed, restratification processes are just as important as the surface stress and buoyancy forcing cited in the paper's introduction (page 1, line 41) in controlling mixed layer depth.

The authors employ the de Boyer Montegut et al. (2004) protocol (at times modified to use a smaller temperature difference criterion) to estimate the depth of the mixed layer depth in CTD data used as ground truth for their acoustic scheme. Adoption of a technique based on temperature is a bit odd for Arctic data since at cold temperatures, density is so strongly controlled by salinity. While I doubt it would change the main conclusions of the paper, it might be worth trying the Holte and Talley (JAOTech, DOI: 10.1175/2009JTECHO543.1) algorithm, particularly for those cases where there was disagreement between the methods. I would also quibble with the exclusion of very shallow mixed layers in the analysis, though I certainly understand the constraints deriving from vessel draft and acoustic blanking period. In the summer Arctic sea ice zone, the upper ocean can be stratified all the way up to the ice-ocean interface. (Drainage of melt ponds is an important summertime stratification mechanism.) The authors show one such example in figure S1 but I worry such stratification is common throughout much of the Arctic in summer, and that observations from a deep-draft vessel will give biased results. Speaking of vessel draft, there is of course the strong possibility that ships disturb the near-surface stratification, introducing yet another source of error.

My final general comment concerns the interpretation of the acoustic observations, exemplified by the sentence on page 2 line 20: "The increased SNR of wideband echosounders have made it possible to map density stratification in the ocean." The authors don't actually invert their acoustic data to estimate the ocean density profile. Rather, it is my understanding that they equate regions of enhanced acoustic backscatter with regions of enhanced vertical density gradients, the one discussed here being the mixed layer base.

I continue with more specific comments/suggestions:

Page 1 line 32-34: I note that light is also a significant factor impacting phytoplankton growth, which can be impacted by MLD and residence time for phytoplankton near the air-sea interface.

Page 1 line 47: the term "temporal sampling frequency" could be confusing - I initially thought of the sampling rate of the CTD instruments, not the time between vertical profiles. Page 2 top: I note that the remote sensing observations from the GRACE satellite mission are indicative of more than surface ocean properties.

Page 2 paragraph starting with line 3: I found it curious that this brief history doesn't begin with echosounding to determine water depth. See http://oceanexplorer.noaa.gov/history/electronic/electronic.html

Figure 1 (and others): I found the quality of the figures in this pdf to be not as crisp as I like. I'm hoping this is just a consequence of the review copy that was made available to me and that the published document will be better (i.e., quality more like the similar figures in the lead author's recent Nature paper).

Page 3 line 21: The authors might wish to temper this phrase: "Together, the SWERUS-C3 and AO2016 expeditions spanned the breadth and depth of the Arctic Ocean..." No observations were obtained in the Canada Basin for example.

Page 4 line 38: "A CTD [profile] was collected..."

[Figure]

Page 5 line 35: this sentence has no real content. Much better to make a technical statement and cite a figure in support.

Figure 2 and those similar: Please give the location and date that these data were collected. In this caption and those similar, panel B should, in my opinion, say CTD profile, not profiles, or CTD-derived temperature and salinity profiles.

Page 6 line 11: "EK80 data [are] available "

Page 8 line 3: might be good to note the different criteria for MLD used by these previous authors.

In summary, I believe that after revision, this work will be suitable for publication in Ocean Science.

---

## Referee Comment (RC2) · Anonymous Referee #2 · 11 Mar 2018

This paper presents an interesting and concise account of an innovative acoustic method to detect with high spatial resolution the depth of the ocean mixed layer, or mixed layer depth (MLD), a quantity that is of interest for a number of practical applications in oceanography. It is shown, using acoustic mapping, in combination with CTD profiles, that reliable estimates of the MLD may be obtained using the former method. The main obstacles to reliable MLD estimates are very shallow MLDs (lower than 10 m), or the existence of excessive biological scatterers, which confuse the vertical distribution of the reflection coefficient, by introducing noise. The paper appears to be scientifically sound, and is clearly written, reporting novel results that are worthy of publication in Ocean Science. There are a few non-critical points (listed below) that I would like to see addressed before I can recommend acceptance. Therefore, at this point I recommend that the paper undergoes minor revisions.

Minor comments

1. Page 1, line 20: "These prerequisites [MLD well-defined and absence of biological scatterers] are often met in the open ocean". Given that the study focuses on the Arctic Ocean, can the authors be sure that this remark is of general applicability, and not limited to that ocean? If not, then the necessary cautions should be noted.

2. Page 1, lines 27-28: "generated by wind stress and buoyancy fluxes at the air-sea interface", and lines 41-42: "The MLD is controlled primarily by surface stress (exerted by wind or sea-ice), buoyancy fluxes (heating/cooling, ice melt/formation, or precipitation/evaporation), and dissipation". In this picture, the effect of waves is missing. It has been established that surface waves, through their interaction with the wind stress and generation of Langmuir circulations, exert a decisive control on MLD growth (e.g. Thorpe, 2004, Ann. Rev. Fluid Mech.). This should be recognized.

3. Page 2, line 26: "ensonified". This word is probably unfamiliar to the readership of Ocean Science. Consider providing its significance on its first mention.

4. Page 3, Figure 1: This figure looks somewhat fuzzy (I am not sure if this only occurs in the version available for review, as that happens in some journals). The green dots (particularly on the yellow track), and especially the blue dots, mentioned in the caption, have very limited visibility. Consider using different colours with a better contrast with the blue background.

5. Page 4, line 13: "attitude", and line 23: "match-filtered": again, this terminology may not be familiar to the readers (it is perhaps over-technical), so provide a clarification of its meaning the first time it appears in the text.

6. Page 4, line 35: "Demer et al.", and page 5, line 2: "Lurton & Leviandier". These parts of the citations should not appear between brackets, as the corresponding references are incorporated in sentences. Please correct.

7. Page 6, caption of figure 2: "Vertical magenta lines". These lines are rather difficult to discern in the blue background. Consider improving this aspect.

8. Page 6, paragraph between lines 20 and 27: The authors note that the criterion for detecting the MLD using CTD of using a temperature variation threshold of 0.2 degrees failed in the Central Arctic Ocean. Can they advance a physical interpretation for this behaviour, i.e., why in the Central Arctic Ocean and not elsewhere?

9. Page 7, figure 3: The horizontal scale of panel b in this figure appears no to be similar to that of panel a, but is not indicated. Please add that information.

10. Page 7, table 2: "rmsd". Not much is said in the text about how this quantity is defined and how it differs from the standard deviations in the two columns to the left. Please add that information.

11. Page 8, lines 19-20: "The acoustic method enables the study of internal waves propagating on the layer interface at the base of the mixed layer". What might generate these waves? Is there a possibility that the MLD measurements could be contaminated by waves generated by the remote interaction between the ship and the density interface at the bottom of the mixed layer (often called pycnocline)? It would be a good idea to discuss this aspect, as it might affect the proposed method in general (although not necessarily in the examples presented here).

12. Page 9, line 11: "splitting/merging of layers". Can the authors be a bit more specific about what physical processes might cause this splitting/merging?

13. Page 10, figure 6: This figure is presented as an example of measurements contaminanted by biological scatterers, which makes it difficult (or even impossible) to reliably determine the MLD using the proposed acoustic method. However, in the reflection coefficient graph shown in figure 6c it is still possible to distinguish the MLD as the depth below which the reflection coefficient starts to have a large variability. I wonder whether it would be still possible to usefully determine the MLD by appropriately exploiting that property?

14. Page 10, line 12: "rosette". This word is not used elsewhere in the manuscript, so

consider replacing it by another, more standard word.

15. Page 10, line 28: "lower success rate in coastal areas". Could this also be related to the greater abundance of biological scatterers in those regions? If yes, please add a comment explaining this.

16. There are a number of figures (S1-S5) referenced in the text (page 6, lines 24-25; page 7, lines 3 and 11-12; page 8, line 44; page 10, lines 20 and 34), but not included in the manuscript. Is this just a referencing problem, or are those figures really omitted, in which case allusions to them would need to be removed, with some detriment to a few justifications in the text?
* * *

---

## Author Comment (AC1) · 25 Apr 2018

As a companion to the lead author's paper published (with some different co-authors) in Nature late last year (DOI:10.1038/s41598-017-15486-3) , Stranne and colleagues present what I would term a technical-demonstration paper showing how the depth of the ocean surface mixed layer may be sensed acoustically. The demonstration is set in the Arctic using data from two icebreaker cruises that sampled in both ice-covered and open-ocean regions at various vessel speeds. While the spin is largely positive, several limitations to the technique are discussed that will constrain where and when the approach will yield scientifically useful information.

Apologies for the cumbersome terminology, but I believe it is important to distinguish between the ocean surface "mixed layer" and the "mixing layer." I consider the latter to be the span that is actively being stirred vertically at the time of observation; it can be (and is frequently) thinner than the mixed layer whose base might mark the maximum depth of turbulent stirring in the past.

*We agree with the reviewer on the terminology.  Here we are following the definition of De Boyer, where the threshold value is chosen as to avoid the shallower mixing layer, caused by diurnal variability. This is now explicitly stated in the revised ms.*

My sense is that the acoustic technique presented by Stranne and colleagues preferentially identifies the base of this deeper, possibly remnant surface mixing layer (in part due to the typically larger vertical gradient at the mixed layer base and its greater depth - at least in the data sets presented that were acquired from a large-draft vessel). Either way, I believe it is important to recognize this distinction and discuss if/how each class of "mixed" layer might be observed acoustically.

*We agree with the reviewer. We have added a brief discussion on the mixed layer definition and how we seem to be imaging the "same" MLD with the acoustic method as we (and De Boyer) derive from CTD data using the threshold method.*

One reason for worrying about (weak) stratification within the surface mixed layer is its possible manifestation of restratification processes including submesoscale instabilities (see Timmermans et al., 2011, doi: 10.1175/JPO-D-11-0125.1). Indeed, restratification processes are just as important as the surface stress and buoyancy forcing cited in the paper's introduction (page 1, line 41) in controlling mixed layer depth.

*This is a good point. In the revised ms we mention "lateral advection" in relation to buoyancy fluxes, with reference to the Timmermans paper, as suggested by the reviewer.*

The authors employ the de Boyer Montegut et al. (2004) protocol (at times modified to use a smaller temperature difference criterion) to estimate the depth of the mixed layer depth in CTD data used as ground truth for their acoustic scheme. Adoption of a technique based on temperature is a bit odd for Arctic data since at cold temperatures, density is so strongly controlled by salinity.

*The density threshold approach presented in the De Boyer paper was tested with close to identical results. We opted to use and display the results from the temperature threshold method, as it is simpler plus there are more temperature data available (in e.g. WOD) than there are salinity data, thus rendering this method more useful in a general sense. Note that the same problems we had with the*

*temperature threshold (we had to adjust it for the central Arctic Ocean) also showed up for the density threshold. This is now stated in the manuscript.*

While I doubt it would change the main conclusions of the paper, it might be worth trying the Holte and Talley (JAOTech, DOI: 10.1175/2009JTECHO543.1) algorithm, particularly for those cases where there was disagreement between the methods

*As is pointed out by the reviewer, the focal point of this paper is on the fact that we can observe the MLD acoustically with a high success rate. As this is mainly a methods paper, the comparison with different protocols for deriving MLD from CTD data is, in our opinion, of secondary importance. We hope that a larger acoustic data set (or a compilation of several acoustic data sets) will be used in the future to study these differences more systematically. Currently, the number of groundtruth CTD stations are far too few for any conclusions to be drawn in this regard.*

I would also quibble with the exclusion of very shallow mixed layers in the analysis, though I certainly understand the constraints deriving from vessel draft and acoustic blanking period.

*We determine the presence of an MLD from CTD data by visually inspecting the profiles (as explicitly stated in the original ms). Note that the visually determined MLD can be shallower than 10 m and for all these, the acoustic method is interpreted as failing (this caused more than 50% of the failures during the SWERUS cruise). This is explained in the second comment to Table 1 (double asterisk). If we were to consider only the failures where the MLD is deeper than 10 m, our statistics would look much more convincing.*

In the summer Arctic sea ice zone, the upper ocean can be stratified all the way up to the ice-ocean interface. (Drainage of melt ponds is an important summertime stratification mechanism.) The authors show one such example in figure S1 but I worry such stratification is common throughout much of the Arctic in summer, and that observations from a deep-draft vessel will give biased results. Speaking of vessel draft, there is of course the strong possibility that ships disturb the near-surface stratification, introducing yet another sourceof error.

*In terms of the acoustic method, the placement of the echosounder transponder on the vessel's hull defines an absolute limit in terms of how close to the sea surface we can make observations. As discussed in the ms, the pulse length puts an additional constraint. This is inevitable.*

*Regarding CTD data – the method of making CTD casts from vessels have been around for many decades, and the accuracy and problems involved with such data acquisition can be found in the literature. In the high Arctic Ocean, with engines turned off, and the vessel drifting with the ice at a typical speed of less than 1 m/s, the risk of the vessel itself interfering with the shallow stratification is limited. CTD operations in high waves can be problematic due to the mixing induced by the CTD rosette (moving up and down through the water column with the waves). This is rarely a problem in the Arctic, however, and most of our CTD data seem to be reliable up to one meter or so from the surface.*

My final general comment concerns the interpretation of the acoustic observations, exemplified by the sentence on page 2 line 20: "The increased SNR of wideband echosounders have made it possible to map density stratification in the ocean." The authors don't actually invert their acoustic data to estimate the ocean density profile. Rather, it is my understanding that they equate regions of enhanced acoustic backscatter with regions of enhanced vertical density gradients, the one discussed here being the mixed layer base.

*This is really a question of subtle semantics. It is true that we are not mapping the actual properties of the stratification, but we are mapping density stratification in the sense that we are observing the location of more or less sharp transitions between water masses of different density, in time and space.*

I continue with more specific comments/suggestions:

Page 1 line 32-34: I note that light is also a significant factor impacting phytoplankton growth, which can be impacted by MLD and residence time for phytoplankton near the air-sea interface.

*Light, oxygen and nutrients place important constraints on the primary production, but these are often indirectly controlled by the MLD, as noted by the reviewer. Here we state that the MLD is one of the main factors controlling the primary production, and we hope that the interested reader will go on to read the two papers that we cite (where other aspects on primary production are discussed in detail).*

Page 1 line 47: the term "temporal sampling frequency" could be confusing - I initially thought of the sampling rate of the CTD instruments, not the time between vertical profiles.

*We agree with the reviewer and we have now deleted the words "temporal sampling".*

Page 2 top: I note that the remote sensing observations from the GRACE satellite mission are indicative of more than surface ocean properties.

*Agreed. We have changed the sentence to "..essentially restricted to near sea-surface properties".*

Page 2 paragraph starting with line 3: I found it curious that this brief history doesn't begin with echosounding to determine water depth. See http://oceanexplorer.noaa.gov/history/electronic/electronic.html

*Here we present a brief summary of acoustic water column mapping specifically.*

Figure 1 (and others): I found the quality of the figures in this pdf to be not as crisp as I like. I'm hoping this is just a consequence of the review copy that was made available to me and that the published document will be better (i.e., quality more like the similar figures in the lead author's recent Nature paper).

*Yes, this is a PDF issue. We guarantee that figure quality will be acceptable in the final version.*

Page 3 line 21: The authors might wish to temper this phrase: "Together, the SWERUSC3 and AO2016 expeditions spanned the breadth and depth of the Arctic Ocean..." No observations were obtained in the Canada Basin for example.

*Actually, 3 of the total 21 CTD stations from the Arctic Ocean 2016 cruise were from within the Canada Basin (see Fig 1). We do agree with the reviewer, however, and have reformulated the sentence to "..spanned much of the breadth and depth…"*

Page 4 line 38: "A CTD [profile] was collected..."

*Fixed*

Page 5 line 35: this sentence has no real content. Much better to make a technical statement and cite a figure in support.

*We are not sure what the reviewer means by this comment. What kind of technical statement are we supposed to make? The figure represents our main result and different aspects of the figure are cited further down in the text.*

Figure 2 and those similar: Please give the location and date that these data were collected. In this caption and those similar, panel B should, in my opinion, say CTD profile, not profiles, or CTD-derived temperature and salinity profiles.

*Fixed*

Page 6 line 11: "EK80 data [are] available "

*Fixed*

Page 8 line 3: might be good to note the different criteria for MLD used by these previous authors.

*Fixed*

In summary, I believe that after revision, this work will be suitable for publication in Ocean Science.

---

## Author Comment (AC2) · 25 Apr 2018

This paper presents an interesting and concise account of an innovative acoustic method to detect with high spatial resolution the depth of the ocean mixed layer, or mixed layer depth (MLD), a quantity that is of interest for a number of practical applications in oceanography. It is shown, using acoustic mapping, in combination with CTD profiles, that reliable estimates of the MLD may be obtained using the former method. The main obstacles to reliable MLD estimates are very shallow MLDs (lower than 10 m), or the existence of excessive biological scatterers, which confuse the vertical distribution of the reflection coefficient, by introducing noise. The paper appears to be scientifically sound, and is clearly written, reporting novel results that are worthy of publication in Ocean Science. There are a few non-critical points (listed below) that I would like to see addressed before I can recommend acceptance. Therefore, at this point I recommend that the paper undergoes minor revisions.

Minor comments

1. Page 1, line 20: "These prerequisites [MLD well-defined and absence of biological scatterers] are often met in the open ocean". Given that the study focuses on the Arctic Ocean, can the authors be sure that this remark is of general applicability, and not limited to that ocean? If not, then the necessary cautions should be noted.

*We do not claim that these prerequisites are always met, but that they are often met. It is widely recognized that productivity is generally higher in coastal waters than in the open ocean, which is also consistent with what we see in our data (there are of course exceptions, for instance along the equator due to upwelling). This notion is supported by the difference between the estimated average primary productivity in the world oceans (~50 g C m$^{-2}$ year$^{-1}$) and the estimated average primary productivity in estuarine waters (~250 g C m$^{-2}$ year$^{-1}$), a factor of five.*

*From "Phytoplankton primary production in the world's estuarine-coastal ecosystems J. E. Cloern, S. Q. Foster and A. E. Kleckner".*

2. Page 1, lines 27-28: "generated by wind stress and buoyancy fluxes at the air-sea interface", and lines 41-42: "The MLD is controlled primarily by surface stress (exerted by wind or sea-ice), buoyancy fluxes (heating/cooling, ice melt/formation, or precipitation/ evaporation), and dissipation". In this picture, the effect of waves is missing. It has been established that surface waves, through their interaction with the wind stress and generation of Langmuir circulations, exert a decisive control on MLD growth (e.g. Thorpe, 2004, Ann. Rev. Fluid Mech.). This should be recognized.

*The primary cause for wave generation is wind stress, so waves and Langmuir circulation can be thought of as integral in the statement "surface stress (exerted by wind or sea-ice)".*

3. Page 2, line 26: "ensonified". This word is probably unfamiliar to the readership of Ocean Science. Consider providing its significance on its first mention.

*We agree with the reviewer and have now added a clarification.*

4. Page 3, Figure 1: This figure looks somewhat fuzzy (I am not sure if this only occurs in the version available for review, as that happens in some journals). The green dots (particularly on the yellow track), and especially the blue dots, mentioned in the caption, have very limited visibility. Consider using different colours with a better contrast withthe blue background.

*Yes, this is a PDF compression problem. We guarantee that the figures will look nicer in the final version. We have also improved the contrast between the different colors.*

5. Page 4, line 13: "attitude", and line 23: "match-filtered": again, this terminology may not be familiar to the readers (it is perhaps over-technical), so provide a clarification of its meaning the first time it appears in the text.

*These terms are only to be found in the methods section. While we agree with the reviewer that they are technical, it is likely that only readers that are specifically interested in these technical aspects of our method would go through these details.*

6. Page 4, line 35: "Demer et al.", and page 5, line 2: "Lurton & Leviandier". These parts of the citations should not appear between brackets, as the corresponding references are incorporated in sentences. Please correct.

*Fixed*

7. Page 6, caption of figure 2: "Vertical magenta lines". These lines are rather difficult to discern in the blue background. Consider improving this aspect.

*In the high resolution version of the figure, the magenta lines are easily discernable.*

8. Page 6, paragraph between lines 20 and 27: The authors note that the criterion for detecting the MLD using CTD of using a temperature variation threshold of 0.2 degrees failed in the Central Arctic Ocean. Can they advance a physical interpretation for this behaviour, i.e., why in the Central Arctic Ocean and not elsewhere?

*The simple explanation is that the temperature gradient between the mixed layer and the water mass beneath it is generally smaller. This is now stated in the text. However, the reason for this difference is a more complicated matter that is well beyond the scope of the present manuscript.*

9. Page 7, figure 3: The horizontal scale of panel b in this figure appears no to be similar to that of panel a, but is not indicated. Please add that information.

*The scale on the x-axis is "CTD observations" with equal distance between each observation. This has now been added.*

10. Page 7, table 2: "rmsd". Not much is said in the text about how this quantity is defined and how it differs from the standard deviations in the two columns to the left. Please add that information.

*The root-mean-square deviation is referring to the deviation between the two methods. The standard deviation represents the variability of the MLD observed within each method. We have now clarified this in the text.*

11. Page 8, lines 19-20: "The acoustic method enables the study of internal waves propagating on the layer interface at the base of the mixed layer". What might generate these waves? Is there a possibility that the MLD measurements could be contaminated by waves generated by the remote interaction between the ship and the density interface at the bottom of the mixed layer (often called pycnocline)? It would be a good idea to discuss this aspect, as it might affect the proposed method in general (although not necessarily in the examples presented here).

*We agree with the reviewer that it is possible that part of what we see, in terms of internal waves, might be generated by the vessel. We now discuss this possibility in the text. This would be a general problem for many ship-based observations, including observations made with CTD and free falling microstructure probes. In this study we settle with the fact that we can observe internal waves.*

12. Page 9, line 11: "splitting/merging of layers". Can the authors be a bit more specific about what physical processes might cause this splitting/merging?

*This is actually an open question, see Stranne et al. 2017.*

13. Page 10, figure 6: This figure is presented as an example of measurements contaminanted by biological scatterers, which makes it difficult (or even impossible) to reliably determine the MLD using the proposed acoustic method. However, in the reflection coefficient graph shown in figure 6c it is still possible to distinguish the MLD as the depth below which the reflection coefficient starts to have a large variability. I wonder whether it would be still possible to usefully determine the MLD by appropriately exploiting that property?

*This might indeed be a possibility. A similar approach has already been established: the gradient criterion method, see for example the De Boyer et al. 2004 paper where they review different methods.*

14. Page 10, line 12: "rosette". This word is not used elsewhere in the manuscript, so consider replacing it by another, more standard word.

*This is, as far as we know, the established term for the steel or aluminum structure on which CTD sensors and bottles are mounted.*

15. Page 10, line 28: "lower success rate in coastal areas". Could this also be related to the greater abundance of biological scatterers in those regions? If yes, please adda comment explaining this.

*Fixed*

16. There are a number of figures (S1-S5) referenced in the text (page 6, lines 24-25; page 7, lines 3 and 11-12; page 8, line 44; page 10, lines 20 and 34), but not included in the manuscript. Is this just a referencing problem, or are those figures really omitted, in which

case allusions to them would need to be removed, with some detriment to a few justifications in the text?

*These figures are included in the Supplementary Information (hence the S in front of the figure number).*

---

## Author Response (AR2)

**Topic Editor Decision: Publish subject to minor revisions (review by editor) (10 May 2018) by Neil Wells**

Comments to the Author:

Both in the abstract and introductory section there is some confusion in terms.

p1 line 13 "exchange of heat, freshwater, and gases "or "exchange of buoyancy and gases" would be more general.

*Authors' comment: We agree, and we have changed the text accordingly*

p1 line 30" buoyancy rather than heat". I understand your mixed layer definition only uses temperature, but it is best to be general in the early part of the paper and in the abstract.

*Authors' comment: We agree, and we have changed the text accordingly*